# Biochemical Properties and Physiological Functions of pLG72: Twenty Years of Investigations

**DOI:** 10.3390/biom12060858

**Published:** 2022-06-20

**Authors:** Giulia Murtas, Loredano Pollegioni, Gianluca Molla, Silvia Sacchi

**Affiliations:** Department of Biotechnology and Life Sciences, University of Insubria, 21100 Varese, Italy; giulia.murtas@uninsubria.it (G.M.); gianluca.molla@uninsubria.it (G.M.); silvia.sacchi@uninsubria.it (S.S.)

**Keywords:** d-serine, NMDA receptor, protein-protein interaction, schizophrenia, Alzheimer’s disease

## Abstract

In 2002, the novel human gene G72 was associated with schizophrenia susceptibility. This gene encodes a small protein of 153 amino acids, named pLG72, which represents a rare case of primate-specific protein. In particular, the rs2391191 single nucleotide polymorphism (resulting in in the R30K substitution) was robustly associated to schizophrenia and bipolar disorder. In this review, we aim to summarize the results of 20 years of biochemical investigations on pLG72. The main known role of pLG72 is related to its ability to bind and inactivate the flavoenzyme d-amino acid oxidase, i.e., the enzyme that controls the catabolism of d-serine, the main NMDA receptor coagonist in the brain. pLG72 was proposed to target the cytosolic form of d-amino acid oxidase for degradation, preserving d-serine and protecting the cell from oxidative stress generated by hydrogen peroxide produced by the flavoenzyme reaction. Anyway, pLG72 seems to play additional roles, such as affecting mitochondrial functions. The level of pLG72 in the human body is still a controversial issue because of its low expression and challenging detection. Anyway, the intriguing hypothesis that pLG72 level in blood could represent a suitable marker of Alzheimer’s disease progression (a suggestion not sufficiently established yet) merits further investigations.

## 1. Introduction

In 2002, the pioneering work of Chumakov and collaborators [1] found a link between schizophrenia (SZ) and a 5-Mb region on chromosome 13 (13q33.2 region) in a case–control study of French-Canadian families by single nucleotide polymorphism (SNP) linkages. Two markers from this region were also found to be associated to SZ in a Russian sample. The new genes *G72* and *G30* were identified: *G72* encodes a small protein of 153 amino acids, named pLG72 or DAOA (i.e., d-amino acid oxidase activator, see below) which represents a rare case of primate-specific protein. No orthologs were identified in rodents or other species and even in ape genome the corresponding open-reading frame differ from human counterpart.

In the aforementioned paper, pLG72 was identified as an interacting partner of human d-amino acid oxidase (hDAAO, EC 1.4.3.3), an FAD-containing enzyme that oxidizes d-amino acids. In the human brain, DAAO specifically degrades d-serine (d-Ser) [2,3]. d-Ser is an endogenous coagonist of the *N*-methyl-d-aspartate type glutamate receptor (NMDAR) [4,5,6] essentially involved in synaptic plasticity, learning, memory, and excitotoxicity.

In past 20 years, the research on pLG72 mainly focused on biochemical (see below for details) and genetic studies (also associated to functional magnetic resonance imaging), as well as on the molecular and behavioral characterization of transgenic mice. Genome-wide association studies linked genetic variations within the *G30/G72* locus with psychiatric conditions, including SZ, bipolar disorder, major depressive disorder (MDD), Alzheimer’s disease (AD), etc. [7,8,9,10,11,12,13,14]. Among the investigated SNPs, the rs2391191 in the *G72* coding region is the one most robustly associated to SZ and bipolar disorder in multiple studies [8,15,16,17,18]. It results in the nonsynonymous R30K substitution in pLG72 and affects the protein properties [19] and the cerebral morphology and functions, especially in schizophrenic affected individuals. This mutation was correlated to changes in total frontal lobe volume, cortex thickness, regional homogeneity in different brain regions and white matter integrity in the corpus callosum [20,21,22,23] and was associated with poorer episodic memory function, both verbal and visual [24,25]. When the genome browser at the UCSC site (https://genome.ucsc.edu/) (accessed on 2 April 2022) is queried for ClinVar Short Variant in *G72*, nine SNPs in the coding region are identified, most resulting in missense mutation (R30K, N42K, V85F, P91S, Q136E, K145R). However, their clinical significance, as well as a specific phenotype, is not provided (with the only exception of Q136E, which is reported as benign).

During the years, pLG72 investigations were mainly aimed at understanding whether this protein could represent a suitable biomarker (i.e., if its expression level is altered under pathological conditions) and a novel therapeutic target for relevant neuropsychiatric disorders.

## 2. *G72/G30* Locus and Transcripts

The *G72/G30* locus emerged late during evolution. In particular, *G72* appears as a rapidly evolved orphan gene, with no detectable orthologues in species other than primates [1], as also reported by Ensembl (https://www.ensembl.org) (accessed on 4 April 2022). Its genomic structure is unusual: it was found in a region enriched in repeat elements, with nearest neighboring known genes located 948 kb upstream and over 2 Mb downstream (a “gene desert”, Figure 1A). Part of *G72* coding sequence and the 5′-flanking region is derived indeed from long terminal repeat elements [26], which are prone to transcriptional silencing. At first *G72* was reported to contains eight exons [27], while later analyses indicated that both *G72* and *G30* encompassed nine exons [28]. In silico prediction did not support the presence of a robust promoter: human transcription factor site clusters were identified about 0.7 kb upstream of the *G72* transcription start site (TSS), but this region lacks other prominent promoter features; a putative functional promoter was found 3.5 kb upstream of the TSS, however an EST database search indicated that unlikely *G72* transcript is extended 3.5 kb upstream [27].

*G72* and *G30* are transcribed in opposite directions, but *G72* only is actively translated following an intricate alternative splicing pattern. Among the several described splicing isoforms the one encoding the small pLG72 protein is the longest open reading frame. Six putative alternative transcript variants are currently present at the NCBI database (https://www.ncbi.nlm.nih.gov) (accessed on 8 April 2022): beside the transcript variant 1 coding the full-length protein isoform 1, the other transcript variants encode shorter isoforms. On the other hand, Ensembl reports four primary transcript variants encoding for protein isoforms: both DAOA-202 and -211 (785 and 1368 bp, respectively) encode the “canonical” full-length pLG72 (153 amino acids), while DAOA-208 (823 bp) and DAOA-201 (1433 bp) encode shorter, alternative protein isoforms of 125 and 82 amino acids in length, respectively. DAOA-202 primary transcript is composed by 6 exons and 5 introns (Figure 1B); the other alternative transcript variants derive from the complete skipping of the short exon 1, an alternative splicing site in exon 2 (resulting in intron 1–2 and exon 2 of different length in DAOA-211 compared to DAOA-208 and -201) and the insertion of intron 5–6 sequence in a long terminal exon largely consisting of 3′-UTR, that occurs in DAOA-211 and DAOA-201, but not in DAOA-208 (Figure 1B). Beside these alternative splicing products, the other reported transcripts (DAOA-203-207, DAOA-209 and -210) are indicated as non-coding and possibly involved in the nonsense mediated decay surveillance pathway.

Differently, the only *G30* transcript reported in the database is the DAOA antisense RNA 1, DAOA-AS1-2019, which is classified as a long non-coding RNA in Ensembl and might provide antisense regulation of the *G72* transcript.

## 3. Biochemical Properties

pLG72 (gene sequences AY138546, K30) was expressed in a recombinant form in *E. coli* cells [29], almost exclusively as inclusion bodies. The protein has been largely recovered through a solubilization and refolding procedure in the presence of the anionic surfactant *N*-lauroylsarcosine under alkaline conditions. A final yield of ~70 mg of protein with a purity > 90% was routinely achieved [29]. The refolded recombinant pLG72 possesses a specific tertiary and secondary structure corresponding to ~73% α-helix, ~2% β-strand, and ~24% random coil (as determined by circular dichroism spectroscopy) and binds hDAAO.

Up to now, no experimental 3D structure of pLG72 is available. Attempts to produce a reliable 3D model of the protein were hampered by the absence of experimental structures of homologous proteins. The first reliable model of pLG72 was produced by homology modelling using the I-Tasser server and was based on the sequence UniProt P59103-1 carrying the R30K substitution (Figure 2A) [30]. The predicted 3D model corresponds to a globular elongated protein with a high content of α-helices (~62%, in good agreement with the experimental value, see above). A year later, an alternative model was proposed based on a hybrid ab initio approach adapted for small proteins rich in α-helices (Figure 2A) [31]. The authors predicted two distinct domains: a non-conserved *N*-terminal domain (ND) and a conserved C-terminal domain (CD). Although pLG72 is too small to be a transmembrane protein, it has been speculated that the ND, which resembles a region of the membrane O-GlcNAc transferase proteins, could help the protein to interact with the surface of intracellular membranes (see below). On the other hand, the CD has been proposed to be involved in the interaction with its physiological partner hDAAO [31]. Interestingly, the two predicted models [30,31] are similar in the overall shape of the protein, in α-helix content, and in relative position of K62 (a residue proposed to have a key role in the interaction with hDAAO, Figure 2A).

On the other hand, the 3D model of pLG72 recently predicted by AlphaFold shows a higher content of α-helices (82%) and an overall shape different from the ones previously inferred (Figure 2A) [33,34]. Anyway, the confidence level of the AlphaFold model prediction was very low, with all the residues showing a predicted local distance difference test (pLDDT) < 60 (and ~58% of the residues with a pLDDT < 50); this, and the observation of a high fraction of disordered-prone regions [30], suggest that pLG72 probably possesses peculiar structural properties.

Notably, molecular docking analysis showed that pLG72 binds, through several hydrophobic interactions, the drug chlorpromazine used in SZ treatment, with an estimated K_d_ ~11 μM, a value close to the experimental one (~5 μM) [30,35,36].

## 4. pLG72 Interactors

The main partner of pLG72 is hDAAO; several genome-wide association and meta-studies reported an association of their genes with SZ and other psychiatric disorders. For a review see [14,37].

In 2002, a clone of hDAAO was isolated from a human brain cDNA library using pLG72 as bait in a yeast two-hybrid system [1]. Biochemical studies suggested that pLG72 is an activator of the flavoenzyme; nevertheless, these measurements were performed on pig DAAO and in the presence of a huge molar excess of pLG72 [1]. Subsequent studies showed that the overexpression of pLG72 in U251 human astrocytoma cells did not influence the activity of endogenous hDAAO on d-proline [38], a d-amino acid which is also a substrate of d-aspartate oxidase, a flavoenzyme paralogous to hDAAO involved in d-aspartate catabolism in the brain [39]. In vitro biochemical studies using purified recombinant proteins confirmed the interaction between pLG72 and hDAAO [35] and failed to reproduce the hDAAO activation results. Gel-permeation chromatography and surface-plasmon resonance experiments demonstrated that pLG72 interacts with hDAAO yielding a 200 kDa complex (made of two pLG72 monomers and two hDAAO homodimers), with an apparent K_d_ value of 2–8 µM [19,35]. The main effect of pLG72 binding to hDAAO was a progressive inactivation of the flavoenzyme due to the alteration of its tertiary structure, while the catalytic efficiency on d-Ser and the affinity for the cofactor were not altered [35,36].

The interaction between pLG72 and hDAAO has been also investigated in U87 cells expressing hDAAO and pLG72: while the overexpression of hDAAO decreased d-Ser cellular levels, the co-expression of pLG72 restored the d-Ser content, confirming that pLG72 acts as a negative effector of the human enzyme activity [35]. Interestingly, upon transient transfection the expression of pLG72 in kidney-like HEK293 cells increased hDAAO activity, while in neuron-like SHSY5Y and in astrocyte-like 1321N1 cells no significant differences were apparent, suggesting an intriguing cell-specific effect of the hDAAO-pLG72 interaction [32]. While pLG72 was initially proposed to be in endoplasmic reticulum/Golgi in transfected cells [1], all the following studies identified the protein in mitochondria only. In details, confocal microscopy and FRET analysis showed that pLG72, located on the cytosolic side of the outer mitochondrial membrane, could interact with the cytosolic neosynthesized hDAAO before its translocation to peroxisomes [40]. This interaction decreases the half-life of hDAAO (t_1/2_ from 60 h to 6 h), suggesting that in presence of its modulator hDAAO is more prone to degradation, as also demonstrated in vitro by limited proteolysis experiments [30,36]. Accordingly, we proposed that pLG72 could target the cytosolic form of hDAAO to the ubiquitin-proteasome system, preserving cytosolic d-Ser and protecting the cells from oxidative stress eventually generated by the accumulation of hydrogen peroxide produced by the flavoenzyme reaction in the cytosol [35,40,41].

The predicted interaction mode between hDAAO and pLG72 (using ZDOCK server) shows that pLG72 interacts with the FAD binding domain of hDAAO forming a dimer interface surface of ~1000 Å^2^, an area that potentially allows the formation of a stable complex in solution. In addition, the residues K62 of pLG72 and T182 of hDAAO (that can be in vitro cross-linked by the bis(sulfosuccinimidyl)suberate reagent are close one to each other (12.2 Å, Figure 2B) [30]. The positively charged surface of pLG72 is well suited to interact with the mostly negatively charged hDAAO surface [31]. According to the hDAAO-pLG72 interaction model proposed in [30], R30 does not seem to interact with the surface of hDAAO: the effect of the substitution could mainly depend on a change of the overall protein conformation rather than to a direct alteration of the interface region (Figure 2B).

Jagannath and colleagues investigated the potential effect of the interaction of pLG72 with hDAAO [32] by mean of Molecular Dynamics (MD) simulations using a pLG72-hDAAO model produced by the patchdock docking server [42] and exploiting information about residues at the pLG72 surface potentially involved in the interaction with hDAAO [43]. Interestingly, in the ensuing model, pLG72 mainly interacts with the FAD binding domain of hDAAO, as already suggested by [30] (Figure 2B). The MD simulations showed that the binding of pLG72 to the hDAAO holoenzyme could increase the flexibility of the flavoprotein thus decreasing its stability and rendering the protein more “misfolded-like”. This effect was not observed when hDAAO apoprotein, instead of the holoenzyme, was used in the simulations [32]. These in silico observations agree with the experimental evidence showing that pLG72 interaction alters and destabilizes the tertiary structure of hDAAO holoprotein [36]. Overall, computational and experimental investigations suggest that, from a structural point of view, pLG72 binding to hDAAO holoenzyme results into a less compact, more flexible, and less stable flavoprotein.

In 2014, extensive interaction analysis using the yeast-two hybrid system and the Lumier-assay on HEK 293, COS and N2a cells identified the mitochondrial methionine sulfoxide S reductase B2 (EC 1.8.4.12, MSRB2) as a specific interactor of pLG72 [44]. MSRB2 is part of the mitochondrial MSR system involved in the elimination of cellular reactive oxygen species (ROS): the up-regulation of pLG72 could alter the function of MSRB2 leading to an increase in mitochondrial oxidative stress, a condition found in schizophrenic patients [45]. Notably, pLG72 was also reported to interact with human d-aspartate oxidase [46].

## 5. Putative Biological Roles of pLG72

There are four main suggestions about the physiological role of pLG72. The first one is the “pLG72–DAAO–NMDAR” link: in vitro and cellular studies demonstrated that the pLG72-hDAAO interaction negatively affects the enzymatic activity and half-life of the flavoenzyme [1,19,32,35,36,37,40]. Based on incontrovertible evidence, it is time now to eliminate the misleading term DAOA from literature and to use the term pLG72 only. We suggested that pLG72 could (transiently) interact with hDAAO on the mitochondrial surface: the binding negatively affects hDAAO activity and, in turn, its ability to degrade d-Ser [2,3], the coagonist required for the NMDAR receptor activation. DAAO-pLG72 interaction links abnormal d-Ser levels and NMDAR dysfunction-related neurological disorders, see below.

A second hypothesis regards the ability of pLG72 to affect “mitochondrial function” by modulating the morphology of the mitochondrial network. The *N*-terminal portion of pLG72 contains a putative mitochondrial trans-location sequence: immunostaining and confocal analysis showed that pLG72 is mainly localized in mitochondria [35,38]. Increased pLG72 protein levels induced mitochondrial fragmentation (increased number of mitochondria/mm of the dendritic tree along the dendritic shafts) without inducing apoptosis in the COS-7 cell line and primary neurons and changed mitochondrial morphology (the vesicular mitochondria were more mobile, possibly delivered more rapidly to sites of intense growth) and dendritic branching in primary hippocampal neurons (within the period of active dendritogenesis only) [38]. Interestingly, pLG72 overexpression affected gene expressions of several mitochondrial-related proteins and increased ROS production in U87 glioblastoma cells, a rise quenched by the general ROS quencher Tempol [47]. The same work reported the co-localization of ectopically expressed pLG72 with superoxide dismutase 1 (SOD1, EC 1.15.1.1), resulting in SOD1 aggregation and partial inactivation, and in decreased cell proliferation. Later in vitro studies on recombinant wild-type pLG72 and its R30K variant excluded the establishment of a stable complex and the alteration of SOD1 activity and stability by pLG72 [48]. When ectopically expressed in glioblastoma U87 cells, pLG72 did not affect cell viability and ROS/superoxide production. Furthermore, pLG72 largely localized to mitochondria and SOD1 was largely cytosolic and the ectopic expression of pLG72 did not alter the expression of SOD1 and its aggregation. Altogether, we excluded a modulation of SOD1 function and aggregation by pLG72. Anyway, pLG72 transgenic mice (G72Tg) show SZ-relevant behaviors, mitochondrial dysfunction, and higher ROS production [49,50]: treatment with the antioxidant N-acetyl cysteine improved the cognitive deficits of these mice. Increasing pLG72 protein expression induces mitochondrial oxidative stress by altering MSRB2 function (which is responsible for the elimination of cellular ROS) [50], this linking ROS production with pLG72, as originally suggested by [51].

The third hypothesis is related to the ability of pLG72 to bind the “flavin mononucleotide (FMN) cofactor” [19,35] and modulate the FMN-containing oxidoreductase activity in the respiratory complex I [50]. G72Tg mice showed reduced activity of aconitase and complex I of the respiratory chain, reduced levels of glutathione, increased oxidative stress and expression of detoxifying glutathione transferases [50]. Furthermore, NMDARs located on the inner membrane of mitochondria are insensitive to glycine as coagonist (this indicating d-Ser as the physiological modulator) and might represent a further conduit for calcium entry in mitochondria [52].

Finally, a very recent multi-omics study revealed “decreased expression of myelin-related proteins” and NAD-dependent protein deacetylase sirtuin-2 (Sirt2), as well as increased expression of the scaffolding presynaptic proteins bassoon (Bsn) and piccolo (Pclo) and the cytoskeletal protein plectin (Plec1) in G72Tg compared to wild-type mice, coupled to a significant decrease of nicotinate levels (a precursor of NAD(P) cofactors) [53]. Hypomyelination has been reported in SZ, as well as synaptic alterations, including changes in presynaptic neurotransmitter release and post-synaptic characteristics. Furthermore, in the rat cerebellum, both Bsn and Pclo were identified as DAAO interacting proteins: Bsn was shown to co-localize with DAAO and to inhibit its activity [54], this further linking pLG72 with DAAO functionality.

## 6. pLG72 Levels in Human Brain

The level of pLG72 in the human body is a controversial issue because its detection is hampered by both low expression and technical issues.

The G72 mRNA and pLG72 expression (the latter detected by Western blot analysis using custom-made antibodies against a pLG72 peptide, then validated on the recombinant protein) was evaluated on human brain samples [27]. No G72 mRNA was identified in adult brain, amygdala, caudate nucleus, fetal brain, spinal cord and testis of both human cell lines and SZ/control postmortem samples. Similarly, pLG72 protein was also undetectable in a number of human brain regions (including cerebellum, amygdala, and frontal cortex), spinal cord and testis. These authors concluded that the protein is not normally present in several human tissues. Anyway, the Human Protein Atlas (https://www.proteinatlas.org/ (accessed on 20 April 2022) reported the detection of pLG72 protein in the cerebellum and cerebral cortex using immunohistochemistry [55].

Later on, the level of G72 (and DAAO) mRNA and protein was studied in six brain regions (i.e., cerebellum, brain stem, amygdala, striatum, thalamus and frontal cortex) in human *postmortem* brain samples (from 16 weeks of gestation to 91 years) [56]. pLG72 protein level was estimated using a commercial enzyme-linked immunosorbent assay (SEJ297Hu, Cloud-Clone Corp.); the specificity of these tests was substantiated by Western blot analysis although no stringent controls were reported. The pLG72 protein was detected in all brain regions (ranging in the 0.01–0.04 µg/mg total proteins level compared to 0.2–0.7 µg/mg total proteins for DAAO) while no G72 mRNA was identified [32,57], suggesting a tightly regulated or extremely localized expression. Methylation appears indeed a relevant mechanism to control G72 gene transcription [57]: the putative promoter region is highly methylated at both CpG and non-CpG sites. The levels of modification at the latter sites is impressive, ranging between 15% and 35% [58]. Notably, an alternative mechanism that would explain the extreme difficulty in detecting pLG72 encoding transcript in the brain relies on the RNA instability motif found in the 5′-UTR of G72 gene [27]. Moreover, numerous miRNA target sites have been predicted in the promoter, the 5′-UTR and the 3′-UTR regions [32], this representing a further mechanism putatively responsible for the negligible levels of pLG72 protein detected in different brain regions.

pLG72 exhibits a region-specific expression pattern in the human brain [32]: its level was higher in the frontal cortex (set to 100%) than in amygdala (80.4%), thalamus (69.6%) and cerebellum (65.2%); in the striatum (set to 100% and showing the comparatively higher level) the protein level was higher than in amygdala (66.7%) and cerebellum (54.1%). This work also detected a significant positive correlation (controlled for age) between hDAAO and pLG72 proteins in all of the brain regions studied except for the frontal cortex.

At the cellular level, pLG72 was detected in neuron-like (SH-SY5Y), astrocyte-like (1321N1) and kidney-like (HEK293) human cell lines [32] by Western blot analysis at the expected size of 18 kDa. pLG72 levels were comparatively higher in HEK293 cells than in 1321N1 and SH-SY5Y cells (absolute values were not reported).

## 7. Blood pLG72 Levels: A Novel Biomarker?

Concerning the presence of pLG72 in serum, its peripheral detection may be a useful surrogate to evaluate the protein level in the central nervous system in case the protein is expressed in both. The investigation of 30 patients affected by SZ (matched with healthy controls by age and gender) reported markedly higher pLG72 levels in the plasma of medicated patients than in healthy controls, as well as in drug-free SZ patients (4–5 vs. 1.2 µg/mL, Table 1) [59].

Later on, a study on healthy controls plasma or cerebrospinal fluid reported not significantly different pLG72 levels in patients with SZ or MDD (~0.3 ng/mL, Table 1) [13]. Notably, the reported values are two orders of magnitude lower than the ones reported by [59]. No correlation was apparent between pLG72 levels and age and between plasma or CSF pLG72 levels and positive symptoms score on the positive and negative syndrome scale (PANSS) in SZ and with depression severity in MDD. By using the same detection method, a statistically significant increase in pLG72 levels was reported in serum of SZ patients vs. healthy controls (496 vs. 346 pg/mL, Table 1) [61]. Notably, also in this study the estimated values were significantly lower than the ones previously detected [59,62]: the absence of published primary data does not allow to provide an explanation for the observed difference. The following study by Lin’s group concluded that pLG72 protein alone, without adding G72 SNPs, may discriminate patients with SZ from healthy individuals [62]. On a total of 149 subjects, the mean pLG72 blood level of SZ patients was markedly higher than that of healthy controls (4.057 ± 2.594 µg/mL vs. 1.147 ± 0.574 µg/mL, Table 1).

Most recently, the plasma level of pLG72 and additional putative biomarkers for SZ susceptibility in the Taiwanese population were studied (Table 1) [63]. The analysis revealed that the average value of AUC (area under the receiver operating characteristic curve) for the ensemble boosting prediction model with random under sampling was 0.9242 ± 0.0652 using five biomarkers including DAAO-, pLG72- and melatonin levels, age, and gender and performed maximally among predictive models to infer the relationship between SZ and biomarkers.

The levels of pLG72 were also evaluated in the peripheral serum of 376 individuals (Han Chinese population) consisting of five groups (healthy elderly, amnestic mild cognitive impairment [MCI], mild, moderate, and severe AD) [60]. This study revealed increased pLG72 levels in the MCI and mild AD groups when compared to the healthy group (2.3 and 2.9 vs. 1.4 ng/mL, respectively, Table 1). However, pLG72 levels in the moderate and severe AD groups were lower than those in the mild AD group (2.7 and 2.0 vs. 2.9 ng/mL). Unluckily, the lack of stringent controls does not allow to consider this intriguing result as incontrovertible evidence of pLG72 presence in blood even because the reported values are three-orders of magnitude lower than previous values [59,62]. Furthermore, d-Ser level and D-/(D+L)-serine ratio were significantly different among the groups: l-Ser levels were correlated with the pLG72 levels. The authors used these results as a support of the hypothesis of NMDAR hypofunction in early-phase dementia and NMDAR hyperfunction in late-phase dementia. However, pLG72 levels in severe AD were reported to be not significantly altered [60], this suggesting that pLG72 may not be suitable as a biomarker for late AD. Moreover, we recently demonstrated that the d-Ser level and D-/(D+L)-serine ratio in AD patients increased with the progression of the disease and were significantly higher than in the healthy control [64,65]. Our study proposed these two parameters as novel and valuable biomarkers for the progression of AD, allowing to discriminate CDR 2 and CDR 1 patients from healthy (CDR 0) individuals (CDR = Clinical Dementia Rating scale). These findings also suggest that pLG72-mediated hDAAO modulation might contribute to NMDAR dysfunction and AD progression.

## 8. Conclusions

20 years of investigations did not shed full light on the role(s) of pLG72. In particular, its low in vivo expression and its difficult detection and quantification (see Table 1) hindered the studies on pLG72. This point asks for additional experimental controls, such as the addition of known amount of recombinant pLG72 in serum samples before Western blot analysis, the depletion of pLG72 before immunoprecipitation analysis, and the use of mass spectrometry to identify peptides arising from pLG72. From a pathological point of view, decreased levels of d-Ser have been correlated to several psychiatric disorders [66], thus compounds able to stabilize the pLG72-hDAAO complex may represent novel therapies to restore d-Ser concentration. On this side, several compounds with a hDAAO inhibitory effect that was more potent (class A compounds) or 5- to 10-fold less potent in the presence of pLG72 (class C compounds) were identified [67]. Finally, the intriguing hypothesis that pLG72 level in blood could represent a suitable marker of AD progression has not been sufficiently established yet: its link with the (de)regulation of brain d-Ser levels merits further investigations.

## Figures and Tables

**Figure 1 biomolecules-12-00858-f001:**
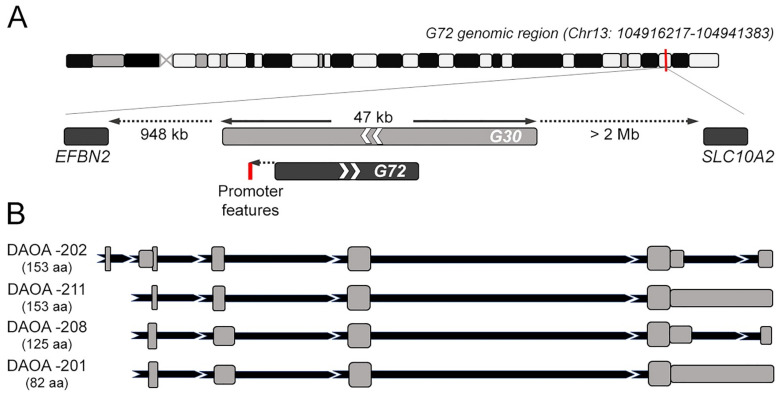
Genomic context and alternative primary transcript variants for *G72* gene. (**A**) Representation of *G72* genomic context on chromosome 13. The overlapping *G30* gene, as well as neighboring genes are shown in the detail. (**B**) Structure of the *G72* alternative transcripts coding for pLG72 protein isoforms as reported in the UCSC Genome Browser and Ensembl. Exons are drawn as grey boxes, intronic sequences are reported in black. DAOA-202, ENST00000375936.9; DAOA-211, ENST00000618629.1; DAOA-208, ENST00000595812.2; DAOA-201, ENST00000329625.9.

**Figure 2 biomolecules-12-00858-f002:**
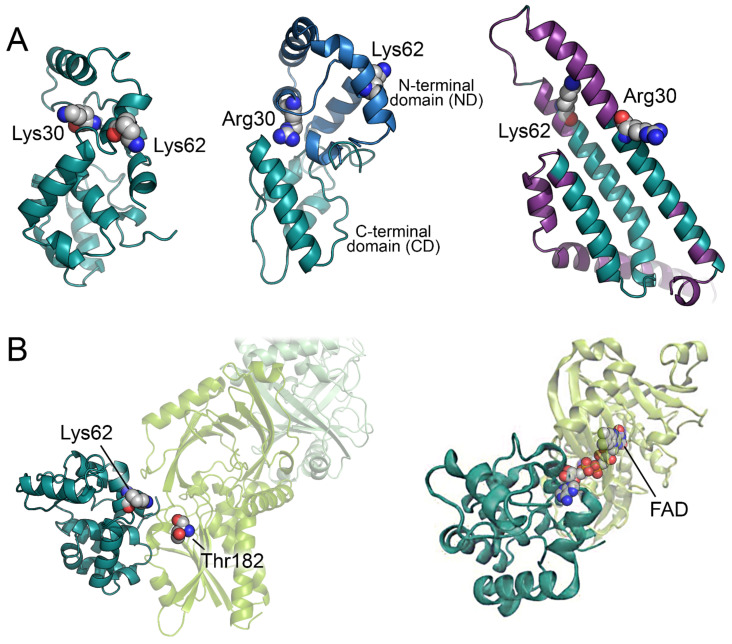
Predicted models of the pLG72 3D structure and of the pLG72-hDAAO complex. (**A**) 3D models of pLG72 as predicted by [30] (**left**), [31] (**center**), and AlphaFold (**right**). In the latter structure, regions with an pLDDT < 50 are depicted in purple. (**B**) Predicted interaction between pLG72 and hDAAO as reported in [30] (**left**) and [32] (**right**). pLG72 and hDAAO are represented as cartoon (teal and pale green, respectively). Significant residues of pLG72 and Thr182 of hDAAO are shown as grey spheres.

**Table 1 biomolecules-12-00858-t001:** pLG72 levels in human plasma of healthy controls and patients affected of MDD, AD and SZ. The number of analyzed samples is reported in parenthesis and the replication group is depicted in italics.

Sample	Healthy Controls	Pathological State			
		MCI (CDR = 0.5)	Mild AD (CDR = 1)	Moderate AD (CDR = 2)	AD (CDR = 3)
Plasma ^a^ ng/mL	1.4 ± 0.7 (108)	1.4 ± 0.7 (81)	2.3 ± 1.1 (124)	2.9 ± 1.6 (35)	1.4 ± 0.7 (28)
		MDD		SZ	
Plasma ^b^ pg/mL	~300 (27) *~415 (30)*	~320 (26)		~280 (27) *~390 (40)*	
CSF ^b^ pg/mL	~ 20.5	~20		~18	
Plasma ^c^ µg/mL	~1.2 (30)			~5 (30) medicated ~4 (27) drug free	
Plasma ^d^ µg/mL	1.147 ± 0.574 (60)			4.057 ± 2.594 (89) unmatched 4.188 ± 2.772 (66) matched	
Serum ^d^ ng/mL	0.346 ± 0.102 (60)			0.496 ± 0.152 (107)	
Plasma ^e^ µg/mL	1.68 ± 0.81 (86)			3.24 ± 1.80 (355)	

CDR, clinical dementia rating. ^a^ pLG72 detection by Western blot using the G72N(15) goat anti-pLG72 antibody (sc-46118, Santa Cruz Biotechnology, Dallas, TX, USA) [60]. ^b^ pLG72 detection was performed by ELISA test (Cusabio Biotech Co. Ltd., Wuhan, China) and the specificity of the antibody used was evaluated by Western blot and by immunoprecipitation with commercial anti-pLG72 antibodies (Cusabio, Houston or Santa Cruz, Dallas, TX, USA) [13]. ^c^ This work established pLG72 levels by Western blot (in duplicate) using the G72(N15): antibody: published data do not show molecular mass standards and the recognized band possessed a different migration compared to control pLG72 [59]. ^d^ Primary data obtained by Western blot analysis using the G72(N15)antibody have not been shown [61,62]. ^e^ [63].

## Data Availability

Not applicable.

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
