# Peer review of "Biochemical Properties and Physiological Functions of pLG72: Twenty Years of Investigations"

_biomolecules, 2022, doi:10.3390/biom12060858_

Round 1

Reviewer 1 Report

Reviewer comments

Summary

Through Ensembl data reports and in silico prediction, the author addressed the pGL72 protein in detail, which approximated the time course of the novel human gene G72. This work makes a timely and crucial contribution to the depiction of pGL72, which is an exclusive and intriguing topic related to major depressive disorder (MDD), Alzheimer’s disease (AD), etc. Nevertheless, overall for a review article, the paper appears to be too short and relatively intricate to be understood by a broad readership. My decision will be prone to acceptance in the context of a more longitudinal description regarding pGL72 based on the time course in chronological order.

General comments:

  1. My main concern is that the article content seems not sufficient yet to understand the pGL72 thoroughly in the introduction section, e.g., the associated correlation between the implicit and explicit introduction of the G72 gene, adding a much more detailed and longitudinal presentation about pGL72 research development process, etc. The horizontal depiction of this paper seems all right.
  2. Given that the title of this manuscript seems to be ambiguous and marginal, the title may need to be revised to reflect more information based on the G72 gene and the entire manuscript context. THX.

Specific comments:

  1. On page 2, with respect to the caption underneath Figure 1, corresponding to line 75, the word “in” got a duplicate. Please correct it, that no duplicate will be found.
  2. On page 3, corresponding to line 104, “with a purity >90%” should be corrected as “with a purity > 90%”, leave a character space between “>” and 90%”, furthermore, also need to check the whole manuscript to amend this issue.
  3. On page 8, regarding Table 1, there are many lines between upper and lower black bold lines. It seems to be not appropriate in terms of universal Tables in the literature. The Table layout could be revised to be more concise and elegant, like Three-line Table style.

Author Response

REFEREE 1:

General comments:

My main concern is that the article content seems not sufficient yet to understand the pGL72 thoroughly in the introduction section, e.g., the associated correlation between the implicit and explicit introduction of the G72 gene, adding a much more detailed and longitudinal presentation about pGL72 research development process, etc. The horizontal depiction of this paper seems all right.

We thank the Referee for his/her comments and criticisms (see also the following points).

We need to mention that the text is at the limit of the length allowed by the journal. Anyway, we introduced a couple of additional sentences to better explain the main lines that were followed by authors in studying pLG72. We hope that the introduced changes will comply with the point arose by the Referee and allow to improve the clarity of the presentation.

Given that the title of this manuscript seems to be ambiguous and marginal, the title may need to be revised to reflect more information based on the G72 gene and the entire manuscript context. THX.

Following the Referee’s criticism, we changed the title.

Specific comments:

On page 2, with respect to the caption underneath Figure 1, corresponding to line 75, the word “in” got a duplicate. Please correct it, that no duplicate will be found.

On page 3, corresponding to line 104, “with a purity >90%”should be corrected as “with a purity > 90%”, leave a character space between “>” and 90%”, furthermore, also need to check the whole manuscript to amend this issue.

Both requested changes have been introduced.

On page 8, regarding Table 1, there are many lines between upper and lower black bold lines. It seems to be not appropriate in terms of universal Tables in the literature. The Table layout could be revised to be more concise and elegant, like Three-line Table style.

We modified the Table following the suggestions (and we eliminated a column). We need to mention that the original formatting of the Table was different from the one reported in the present version: we think that changes are due to the journal’s style.

Reviewer 2 Report

This really interesting and comprehensive review by Murtas and colleagues presents and discusses current state-of-art of small protein pLG72 molecular biology and physiology in the context of its possible role in the cellular mechanisms of Alzheimer disease pathology.  Presented article seems to be valuable for all specialists working in the field of clinical neurology and pharmacology. It may potentially help to introduce some novel strategies in the diagnostics and therapy of some neurodegenerative diseases. Finally, authors suggest that pLG72 may be considered a new marker of AD progression. A detailed characteristics of pLG72 genetics and neurochemistry both in animal and human brain is complete and very informative for the readers. Figures (especially predicted 3D molecular models) and tables are also well designed and clear. To sum up, this review can be perceived as important and original contribution to the field of neuroscience.

Author Response

REFEREE 2:

We thank the Referee for his/her comments.

Reviewer 3 Report

This is an excellent review.

Some minor points are:

Do the authors see a therapeutic strategy evolving here regarding potential pLG72 analogues and perhaps blockers to understand resulting pathophysiology?

Abstract

Should the sentence piece “pLG72 was proposed to target the cytosolic form of D-amino acid oxidase to the degradation,….” be “pLG72 was proposed to target the cytosolic form of D-amino acid oxidase for degradation,…” ??

Body of Text

The authors state that “Based on incontrovertible evidence, it is time now to eliminate the misleading term DAOA from literature.” What do they propose to call it?

Body of Text

Are the actions of pLG72 confined to within the cells it is generated or do you see it as a trans-cellular communicator?....see next point

Body of Text

Is there any information on the exact intracellular location of pLG72?...vesicles for exocytosis to affect neighboring cells for example?

Author Response

REFEREE 3:

Some minor points are:

Do the authors see a therapeutic strategy evolving here regarding potential pLG72 analogues and perhaps blockers to understand resulting pathophysiology?

This is an interesting suggestion! For sake of honesty, we’d mention that we did not take into consideration this opportunity. We mentioned the third generation D-amino acid oxidase inhibitors that are affected by pLG72 presence (see Conclusions section).

Abstract

Should the sentence piece “pLG72 was proposed to target the cytosolic form of D-amino acid oxidase to the degradation,….” be“pLG72 was proposed to target the cytosolic form of D-aminoacid oxidase for degradation,…” ??

The change has been introduced.

The authors state that “Based on incontrovertible evidence, it is time now to eliminate the misleading term DAOA from literature. ”What do they propose to call it?

We propose to use pLG72 acronym. This is now clearly stated at page 7.

Are the actions of pLG72 confined to within the cells it is generated or do you see it as a trans-cellular communicator?....see next point

Is there any information on the exact intracellular location of pLG72?...vesicles for exocytosis to affect neighboring cells for example?

This is an interesting point that, to the best of our knowledge, was never investigated. We now stated at page 6 that “While pLG72 was initially proposed to be in endoplasmic reticulum/Golgi in transfected cells [1], all the following studies identified the protein in mitochondria only.”

Round 2

Reviewer 1 Report

As per the author’s careful, responsible, and indispensable responses in the entire manuscript, my decision is prone to accept it after my check one by one.